# Accentuated Peripheral Blood NK Cytotoxicity Forms an Unfavorable Background for Embryo Implantation and Gestation

**DOI:** 10.3390/diagnostics12040908

**Published:** 2022-04-06

**Authors:** Boris Dons’koi, Oksana Onyshchuk, Iryna Kononenko, Vira Sirenko, Natalia Bodnar, Andrii Serbyn, Anzhela Kozachok, Yulia Brovarska, Dariia Osypchuk, Yaroslava Anochko, Viktor Chernychov

**Affiliations:** 1Laboratory of Immunology, State Institution (Institute of Pediatrics, Obstetrics and Gynecology NAMED AFTER ACADEMICIAN O. Lukyanova of the National Academy of Medical Sciences of Ukraine), 04050 Kyiv, Ukraine; dariia.osypchuk@gmail.com (D.O.); yasia-anoshko@ukr.net (Y.A.); chernyshov@ukr.net (V.C.); 2Reproductive Medicine Clinic “DAHNO IRM”, 04050 Kyiv, Ukraine; oksana.uarm@gmail.com (O.O.); ikonon@ukr.net (I.K.); vera.sirenko@gmail.com (V.S.); fullife.n@gmail.com (N.B.); andrey.serbyn@gmail.com (A.S.); anzhelina01@gmail.com (A.K.); brovarskaya@ukr.net (Y.B.)

**Keywords:** NK lymphocytes, NK cytotoxicity, IVF, embryo implantation, pregnancy failures, preimplantation genetic diagnosis

## Abstract

**Problem** Many studies have demonstrated the negative impact of high rates of NK cytotoxicity (NKc) on reproductive processes, but there is no agreement as to the appropriateness and feasibility of using the NKc for routine diagnostic in IVF patients. This study conducted a retrospective analysis of embryo transfer (ET) success rates and live birth rates (LBR) in patients with different NKc values. **Method of study** 1854 cycles of ET were selected and divided into three groups according to NKc levels, and randomized by anamnesis and age: normal (nNKc, *n* = 871), elevated (eNKc, *n* = 759), and reduced NKc (rNKc, *n* = 123). ET with donors’ embryos (*n* = 101) were analyzed separately. NKc-to-K562 was measured in PBMC (peripheral blood mononuclear cells) by flow cytometry before ET. The patients did not obtain any additional treatments. **Results** Patients with eNKc, in addition to having reduced clinical pregnancy rates (OR1.59, *p* < 0.0001), had increased levels of subsequent pregnancy failures (OR2.545, *p* < 0.0001) when compared to nNKc patients. As a result, patients with eNKc had almost half the LBR than patients with nNKc (OR2.2, *p* < 0.0001). In patients with rNKc, LBR was also lowered. eNKc was equally unfavorable for implantation and delivery in cryo- or fresh cycles. Markedly, eNKc was much more unfavorable for reproduction than slightly elevated NKc. The donor’s embryos were implanted irrespective of the recipient’s NKc levels, but the later stages of pregnancy were worse in patients with eNKc. **Conclusions** Our findings highlighted the negative impact of high levels of NK cytotoxicity on pregnancy outcomes.

## 1. Introduction

NK cells play a critical but poorly understood role in reproductive processes such as implantation, trophoblast invasion, and spiral artery remodeling [1,2]. On one hand, elevated NK cell frequencies [3,4], cytotoxicity [5,6], expression of activating ligands [7], and imbalance between inhibitory and KIR receptors expression on NK cells [8] are associated with different reproductive failures. A strong association was found between peripheral blood NK cell numbers and infertility. Pooling studies showed significantly higher NK cell numbers in infertile women than fertile controls [9]. On the other hand, NK cell activity is also clearly necessary for reproduction. It was shown that fetal growth restriction is associated with reduced proportions of NK in the endometrium and decidua [10] and that NK cell depletion in mice leads to placental abnormalities, thickening of the spiral artery [11], and fetal loss. This nature of NK lymphocytes explains why IVIG therapy becomes truly effective when administered to patients with excessive NK levels or activity [12].

NK cytotoxic reaction, measured by the NK assay against HLA-negative K562 cell lines, is “gold standard” for assessment of NK activity in vitro [13]. It is independent of individuals’ KIR-HLA combination, does not reflect individuals’ “KIR education”, and represents the basic virtual physiological NK capacity for killing [14].

We have been investigating the association between reproductive outcome and NK cytotoxicity since 2006 and have conducted >20,000 NKc tests as routine diagnostics. This study conducted a retrospective analysis of embryo transfer success rates in patients with different NK cytotoxicity values.

## 2. Material and Methods

### 2.1. Blood Sample Collection

NK cytotoxicity was routinely investigated in patients who had at least one incident of idiopathic failures in previous IVF cycles (2006 to 2019). In the overall analysis, we took more than 8000 determinations of NK cytotoxicity in more than 4000 patients from the “DAHNO Institute of Reproductive Medicine”. The clinic is adjacent to a laboratory (800 m) and guarantees blood samples’ transportation at optimal times and conditions. Amounts of 5 mL peripheral blood were collected from 9 a.m. till 12 p.m. and delivered to the laboratory no later than 1 p.m. Thus, no more than 5 h passed after the blood collection and onset of lymphocyte isolation. Effectors (peripheral blood mononuclear cells PBMc) were isolated on a Histopaque-1077 density gradient (“Sigma”, Burlington, MA, USA). Isolated effector population contained <5% granulocytes and 5–12% monocytes.

### 2.2. Study Design

We select clinical history of 2100 ET cycles for retrospective analysis from more than 8000 ET cycles. For a pre-randomized study, 2100 cycles of ET were selected according to the criteria: the ET was performed no more than one year after the last NKc examination (at least one good quality embryo was transferred); patients who were younger than 39 were negative for anti-cardiolipin antibodies, with normal karyotype, and did not have insufficient endometrium and thrombotic disorders. Patients did not obtain any additional treatment. In the case of high or low NKc results, the determination was repeated in most cases. The patients were divided into three groups according to the NKc levels and randomized by age as described below.

We analyzed implantation rate (IR) per ET as a part of hCG positive cases per ET (>50 u/mL on the 14th day after ET) (biochemical, ectopic, and anembryonic pregnancies were included in IR and annualized as separate groups and not included in clinical pregnancy rate calculation). Biochemical pregnancies were calculated as hCG-positive cases that were not confirmed as an egg in the uterus by US on the fifth week.

Clinical pregnancy rate (CPR) was calculated as percentage of the ongoing pregnancy (gestational sac in uterus by US on 5 week.) per ET. Pregnancy failures were calculated as a part of the failed pregnancy from the confirmed ongoing pregnancy.

Live birth rate (LBR) were the number of births of a live baby after 24 w/g per ET. We did not obtain clinical information about subsequent pregnancies ongoing in some patients.

We excluded them from the calculation of PR or LBR.

### 2.3. Randomization of Groups

At “pre-randomization” retrospective analysis, we have seen significantly worse IVF success rates in patients with low and high NKc than those who had normal NKc. However, patients with high NKc had a significantly higher number of previous unsuccessful cycles (mean of 3.1) when compared to (1.95) patients with normal NKc. At the same time, patients with high NKc were significantly older and had reduced average AMH levels than patients with normal NKc. To randomize these two groups, we removed from the high NKc and normal NKc groups all patients who had 5 or more unsuccessful history attempts (*n* = 145). In this way, we were able to identify (and compare), in both groups, the age and severity of patients’ reproductive history. Other parameters (number of fresh or cryo-cycles, endometrial thickness, number of embryos transferred, AMH, FSH, LH, testosterone, TSH hormone levels, DHEA,17 -OP) also did not differ significantly between the groups, as seen in Table 1. Reduced NKc group remained un-randomized by the number of previous unsuccessful attempts, even after exclusion of all patients who had 5 or more unsuccessful IVF attempts. We divided patients into 3 groups according to the NKc levels; normal (nNKc, *n* = 871), having elevated (eNKc) (*n* = 759), and having reduced NKc (rNKc *n* = 123). We analyzed the success rates in the investigated ET.

Groups of patients with donor embryo transfer were analyzed separately (nNKc, *n* = 51) and (eNKc) (*n* = 50). Donor oocytes were obtained from young (up to 34) fertile women who had one or more healthy children born.

In the isolated analysis, we separated patients by age. We calculated average levels of NKc and the frequency of increased NKc in patients 20–30, 30–35, and 35–39 years old. We also analyzed associations of NKc levels with patients’ BMI. We separated the patients by BMI into 3 groups (BMI < 20, BMI 20–30, and BMI > 30).

### 2.4. Determination of NK Cytotoxicity

NK cell activity was measured as described previously [15,16]. Target cells K562 (NK-sensitive, HLA-negative, chronic myelogenous leukemia) were labeled with 5 μM CellTracker™Green CMFDA (5-chloromethylfluorescein diacetate), (Molecular Probes, Eugene, OR USA) and Calcein-AM (O,O′-diacetate tetrakis(acetoxymethyl) ester) (Sigma Aldrich) for 20 min at 37 °C in a humidified 5% CO_2_ incubator. The labeled cells were washed twice in PBS, resuspended in RPMI1640 with 10% NBCS, and counted using Flow-Count™ Fluorospheres (Beckman Coulter, Brea, CA, USA). The effector cells (isolated on a Histopaque-1077 density gradient PBMC (“Sigma”, Burlington, MA, USA) were co-incubated at the effector/target ratios of 30/1, 15/1, and 7.5/1 for 2.5 h at 37 °C in the atmosphere of 5% CO_2_ in the air. After the incubation period, the cells were mixed with 10 μL of PI solution 2 mg/mL (SIGMA) in PBS to stain dead cells. For each E/T ratio, the NK cytotoxicity was measured by analyzing 10,000 target cells/samples using a FACScan flow cytometer (BD Bioscience, San Jose, CA, USA) equipped with CellQuest software In Stat version 3.0 for Windows Graph Pad Software Inc., San Diego, CA, USA.

For each E/T ratio, specific NK cytotoxicity was calculated as a percentage of PI-positive (dead cells) cells minus the level of spontaneous lysis. After, we constructed a calibration curve and calculated virtual cytotoxicity under the condition of E/T ratio (E/T ratios 10/1 and 20/1).

The level of spontaneous lysis was determined as the percentage of dead cells under the same incubation conditions, but in the absence of effector cells. Less than 5% spontaneous lysis of target cells was observed in these experiments (Appendix A).

To correctly measure the E/T ratio, we have labeled permeabilized by solution (0.25% Triton and 1% paraformaldehyde) PBLs with PI, and then added labeled K562 cells. The ratio of E/T, in each sample, was measured by flow cytometry (PBMC–FL3, K562 cells–FL1.) (Appendix A).

For target-gate correction (FL1 lost), we used (50% permeabilized K562 culture) 100μL target cells permeabilized by 50 μL 96% ethanol for 10 s. After the vortexing, 100 μL of PBS and 100 μL of unaffected targets were added. The obtained suspension consists of 50% live and 50% permeable cells (Appendix A).

As the concept of “normal cytotoxicity,” the reference values of NK cytotoxicity favorable for reproductive prognosis was accepted in accordance with our previous work [6,17].

The cytotoxicity activities >30% (E/T ratio 10/1) and <10% (E/T ratio 10/1) were considered as being high and low, respectively. Later, these levels were clinically confirmed in the prospective cohort study [16]. In addition, the same “normal cytotoxicity levels” was defined as the level where no correlation between NK% and NKc existed [18].

### 2.5. Statistical Analysis

The statistical analysis of the results was performed using Fisher’s Exact Test (unpaired, non-parametric, two-sided *p* value) and the Spearman and Pearson correlations (In Stat version 3.0 for Windows Graph Pad Software Inc., San Diego, CA, USA).

## 3. Results

### 3.1. Pregnancy and Live Birth Rates in Patients with Normal and Accentuated NK Cytotoxicity

All clinical information on the onset and the course of pregnancy in the studied patients is summarized in Table 2. In patients with reduced NKc (rNKc) and elevated NKc (eNKc), positive pregnancy tests (hCG > 50 u/mL) were significantly downregulated when compared with patients who had normal NKc (nNKc) levels (OR = 1.467, *p* = 0.0003 for eNKc and OR 1.57 *p* = 0.0369 for rNKc group) (Table 3). Patients with eNKc levels, in addition to having reduced clinical pregnancy rates (OR 1.59, *p* < 0.0001), had increased levels of subsequent pregnancy failures (PF) (OR 2.545, *p* < 0.0001) when compared to nNKc patients. As a result, patients with eNKc levels had almost half the live birth rate per ET (LBR) than patients with nNKc (OR 2.2, *p* < 0.0001). In patients with eNKc, higher frequencies of biochemical pregnancy (OR 1.613 *p* = 0.0635), anembryonic pregnancy (OR 5.934 *p* = 0.0012), and ectopic pregnancy (OR = 5.156, *p* = 0.0571) were registered when compared to nNKc (Table 2).

In contrast, patients with rNKc had almost no biochemical pregnancies nor ectopic pregnancies. In fact, the level of early implantation failures in this group was even lower than in patients with nNKc. Therefore, clinical pregnancy rates in this group did not differ significantly from patients with normal NKc. However, the pregnancy loss rate was slightly higher in this group, leading to a lower live birth rate (OR = 1.642, *p* = 0.06) when compared with nNKc patients (Figure 1).

Thus, accentuated increased NKc negatively affect both the implantation and the subsequent course of pregnancy.

### 3.2. Degree of NKc Elevation

Patients with eNKc had quite different levels of NKc elevation. We divided them into 3 groups-by the degree of NKc and compared the clinical success rates of implantation and pregnancy in these groups. Thus, patients with slightly elevated NKc had not-significantly reduced IR (37.7%) and CPR (30.8%) but still significantly reduced LBR (17.6%, OR 1.463, *p* = 0.0438) compared with the nNKc group where IR, CPR and LBR were 39.4%, 34.3% and 23.8%. Patients with moderately high NKc had a significant decrease in all success parameters IR (30.4%, OR 1.48, *p* = 0.01) CPR (23.6%, OR 1.68, *p* = 0.0013) and LBR (14.2%, OR 1.92, *p* = 0.0007). Among patients with markedly high NKc, success rates were the worst (IR 26.9%, OR 1.761, *p* = 0.0002) CPR (20.6%, OR 2.016, *p* < 0.0001) LBR (6.9%, OR 4.219, *p* < 0.0001) compared to nNKc patients’ rates (Figure 1).

The adverse effect of NKc depends on the degree of accentuation and accumulates with theincrease of NKc levels. But even a slight rise in NKc levels leads to a decrease in the LBR. On the other hand, extremely high NKc in some patients does not interfere with the childbirth.

### 3.3. Type of ET Cycles

We compared patients with eNKc and nNKc depending on the type of cycle (fresh/cryo) and in patients with donated embryos transferred. The cycle type barely affect IVF success. Thus, patients with eNKc had the same reduced CPR and LBR in both “fresh” and “cryo” cycles compared to nNKc patients. (Figure 2).

However, the CPR in patients with donor’s ET with eNKc was as high (19/50) as in the nNKc group with donor’s ET (19/51). However, the LBR in patients with eNKc was still lower (20%) than that of the nNKc (35.4%) (not significant) due to a significantly increased rate of pregnancy failures 35.7% (5/14) in eNKc, being only 5.6% (1/18) in nNKc patients (OR 9.4, *p* = 0.0636).

NK cytotoxicity was equally unfavorable for implantation and delivery in cryo- or fresh cycles. The donor’s embryos were implanted irrespective of the recipient’s NKc level, but the further course of pregnancy was worse in patients with elevated NKc.

### 3.4. Age and BMI

The average levels of NKc and frequency of elevated NKc were similar in all age groups. We have not found any association of high-frequency eNKc cases with BMI levels. Patients with BMIs of <20 had hardly a higher frequency of elevated NKc than in normal BMI groups (20–30) (*p* = 0.08 OR1.39). The average levels of NKc and frequency of elevated NKc were similar in the obesity group when compared to normal BMI groups (20–30). In the investigated population, only 3.3% had BMI > 31.

## 4. Discussion

Individuals with elevated NK lymphocyte levels and NK cytotoxicity have lower risk and better clinical outcomes in the case of viral infections and oncology diseases than patients with lower NK% and NKc levels [19,20,21,22,23]. High efficacy of self HLA recognition by KIR receptors of NK lymphocytes increase the resistance to infection as well as the rate of reproductive problems [24,25].

This work has demonstrated the negative impact of high levels of NK cytotoxicity on the reproductive process efficiency in a large group of “typical IVF patients”. Patients with elevated NKc had worse implantation rates, high levels of biochemical, ectopic, and anembryonic pregnancies, and more pregnancy failures than patients with normal NKc. Markedly high NK cytotoxicity is much more unfavorable for reproduction than slightly elevated NKc. However, implantation of a donor’s embryos with higher implantation–invasion capacities and a lower chance of chromosomal abnormalities and genetic complications did not depend on NKc levels. Donors’ embryo implantation and pregnancy rates were the same in patients with high and normal NKc. However, even in patients with donors’ embryos in pregnancy, the incidence of pregnancy failures among patients with elevated NKc was much higher, and the birth rate decreased. One may speculate that elevated NKc combined with certain genetic and physiological features or embryo deviation might exacerbate the negative pressure on implantation and amplify the adverse effect already at the pre-implantation and early implantation stages. While for highly implantable donor’s embryos, this disadvantage is realized later. The level of NKc does not impact the implantation of donor embryos. In the elevated NKc group, we did not find any pregnancy with congenital abnormalities, but found three in the normal NKc group. In some studies [26], it has been shown that the “normality” of clinical unfavorable NK levels depends on the age of the patients. So, in young patients with “better embryos”, the negative clinically significant levels of NK are even higher than in older patients with “poor embryos”. Relatively lower levels of NK start to be unfavorable in older patients. Kwak Kim and colleagues [27,28] showed that elevated NKc leads to incorrect remodeling of the spiral arteries and subsequent pregnancy complications. We demonstrated this association as well [29]. It explained significant first and early second trimester pregnancy failure rates in patients with their own and donors’ embryos, being at the time when peripheral blood NK cells have the first contact with fetal cytotrophoblast cells in a spiral artery.

It is also possible that donor embryos are not associated with significant implantation problems in patients with a history of implantation failures because of the new HLA/KIR combination variant. It was shown that embryo HLA-C phenotypes could form unfavorable combinations with the maternal KIR/HLA system [30,31]. In patients with multiple implantation failures, the probability that their own embryos will have the same unfavorable combination is higher than in the case of a donor’s embryos.

Flow cytometry enumeration of NK lymphocytes is much simpler, and is used in many investigations. Unfortunately, the determination of the number of NK lymphocytes is not enough to evaluate their functional status [16,18] and seems to have limited clinical significance [32]. It is hoped that the identifying of NK phenotypes and NK subset structures may be more likely to assess NK function, but right now, the NK cytotoxicity test remains the golden standard [33].

Methodological complexity associated with NKc test condition standardization, the short shelf life of the blood samples, and the high cost of the study make it difficult to perform qualitative clinical studies. In the actual investigation, we carefully controlled the transfer of blood samples and used one NKc test protocol design for all inquiries. We used one target line, K562, obtained from ATCC, and a control level of K562 viability of >95%. We also developed original gate control and original E/T calculation methods for more accurate and reproducible test quality. In contrast to commercial testing of NKc (500 US dollars) [34], the cost of one NKc examination in our investigation protocol was on average $20 USD, generally due to the blood sample logistics (once a week and 40–60 samples per investigation).

Critical obstruction of NKc testing in diagnostic protocol is its “Normal reference ranges”. The question remains: what are the normal (optimum) levels of NK, and what is optimal? An additional layer of complexity is that NK% and NKc are dependent on gender, ethnic, and racial backgrounds [35,36,37,38]. Previously, we investigated IVF patients and obtained “clinically favorable” NKc ranges associated with better implantation and pregnancy outcomes [6,17]. In this work, we have demonstrated for the first time the impairment of implanting and pregnancy ongoing in patients with low levels of NKc. Abnormal uterine NK cell numbers and activity have been associated with human and mice gestational complications, including recurrent spontaneous abortion, placental dysfunction, and placenta accrete [39,40,41,42]. It was shown that only 47% of recurrent pregnancy failures displayed uNK values similar to that of controls, 22% had low amounts (<90 cells/mm^2^), and 32% presented elevated numbers of CD56+ uterine NK cells (>300 cells/mm^2^) [43]. So, both elevated and decreased NKc accentuations are unfavorable. We have previously seen that a decline in NK activity and an increase in it may be associated with a poor prognosis [7,44].

Patients with low and high NK CD8 expression and CD158a expression had a reduced rate of pregnancy onset when compared with patients with balanced expression levels [45,46]. This is entirely consistent with the “theory of immune accentuation” [47] because it indicates the presence of an optimal zone, that is, a zone where effective regulation of immune function is possible. In this zone, NK cytotoxicity does not correlate with the NK number [16,18]. The absence of correlation between NK% and NK cytotoxicity was found in the moderate-to-normal zone of NK frequency. This could mean that in these range of NK cells counts, cytotoxicity is not directly determined by NK numbers. Instead, NK lymphocytes counts in the moderate-to-normal zone are sufficient and are compatible with the generation of any NKc levels. We show that the zone where NKc does not correlate with NK frequency is identical to NKc levels favorable for reproducing “normal levels of NKc”, defined in the previous clinical study.

“Normal” level is a favorable value of the parameter where the function can be generated depending on the actual needs of the organism. So, the function is independent of the amount of NK. In contrast, in accentuated NKc zones, both elevated and reduced, NKc starts to correlate significantly with NK frequency. Thus, the function of NK is directly determined by NK levels. The same levels start to be unfavorable for reproduction.

Accentuation of NK through inflammation–affectation or complicated KIR-HLA combination results in decreased/increased NK activity and deregulation of the clonal population balance. Affectation of regulatory mechanisms results in the direct impact of NK frequency on cytotoxicity. Thus, quality directly determines functional quantity. However, for a successful implantation and pregnancy, ongoing NK function needs to be regulated and be able to adapt for the embryo. The accentuated idea is similar to the situation that was described in the study reporting the KIR-HLA recognition model [25]. The study states that “when responding to any specific challenge, an individual with a narrow range of possible NK cell behaviors that is entirely optimized for that challenge will do better than an individual with a wider range of possible NK cell behaviors. However, the individual with narrow NK cell responses will, of course, perform badly against any challenge that requires NK cell responses outside of its range” [25].

Embryo implantation is regulated by fetal–endometrial NK recognition. However, in physiological/fertile conditions, endometrial NK population and peripheral blood NK cells have a different phenotype and function. Peripheral blood NK cell numbers do not correlate with endometrial NK frequency [48,49]. How do NK phenotypes or cytotoxicity in blood represent local endometrium conditions? Additionally, how does this affect embryo implantation? Previously we showed that NK and T lymphocyte populations in the endometrium are tissue autonomic but not independent from the general immune condition. Expression of CD8aa and p46 (CD335) in blood and endometrial NK correlate significantly [49,50]. We find that p46 expression on peripheral blood NK reflects their cytotoxicity status [16]. It also was shown that the peripheral blood NK cell levels reflect changes in decidual NK cell phenotypes in RPL patients. The number of decidual CD16(+) cells was significantly higher in women with elevated peripheral blood NK levels [51]. At the same time, the number of endometrial CD16(+) NK cells was significantly higher in women with RPL [43]. It also was shown that peripheral blood NK cells levels and phenotype reflect decidual NK cell changes during miscarriage [8].

It seems that systemic immune deregulation results in both accentuated NK phenotype/function in peripheral blood and an affectation of endometrial NK receptivity.

There were no congenital abnormalities in fetuses from patients with elevated NKc, which may indicate an over-selectivity of endometrium in patients with elevated NKc. In patients with elevated NKc, none of the implanted embryos formed congenital abnormalities, with a significant number of embryos escaping into the tubes or becoming anembryonic, and with a considerable amount of pregnancy failures. In patients with eNKc, we found four incidences of triplet pregnancy after transfer of two (120 h) embryos from 246 pregnancies in these groups. Additionally, only one incidence of the same triplets from twins’ pregnancy per 314 pregnancies in nNKc patients was observed.

Accentuated NKc forms independent unfavorable conditions for reproduction. Its contrariety was realized equally during the stimulated cycle and for the transfer of the frozen embryo. That is, hormonal stimulation neither worsens, however, nor improves the unfavorable implantation state of eNKc. It is possible that accentuated NKc amplifies the negative effect of the embryo imperfection or complicated KIR/HLA mother/embryo combinations.

This retrospective study is important for understanding the practical possibilities of using NK cytotoxicity as a laboratory method for diagnosing reproductive loss, because this study was conducted on a large population of “typical IVF patients”. “Typical IVF patients” are average IVF patients, not specially selected patients with multiple idiomatic failures or with a specific medical history/diagnosis. However, there are advantages and disadvantages to this study.

The desire to do research on large populations has led to the fact that the study was being performed for many years, and during this time the protocols of stimulation, drugs, and methods of freezing embryos changed. Only a small group of patients received embryo transfer (ET) by pre-implantation genetic diagnosis (PGD)-tested embryos, although it is known that embryonic aneuploidies make a very large contribution to the success of IVF. Therefore, in the future, it is necessary to continue research on patients with PGD-tested, vitrified embryos, which will allow us to investigate the effect of NK cytotoxicity on this group.

A possible limitation of the study is the high mono-ethnicity of the study population. Thus, more than 90% of the studied population was the local Slavic population. HLA-C1/C2 and KIR are insufficiently studied by distribution among the local Slavic population, so it can represent only certain variants of the relationship of these combinations. It is possible that with other dominant combinations, the value of NK cytotoxicity will change both to a greater or lesser extent.

Also, a weak point of the study are the results on hormone levels, because these results were obtained from different laboratories and measured by different methods with different reference values. This could reduce the statistical significance of these parameters.

On the other hand, an important advantage of the study, which may have helped to identify a highly reliable relationship between the accentuated parameters of NK cytotoxicity and the results of the reproductive process, is the study scheme: one IVF clinic/one laboratory/one protocol for measuring NK cytotoxicity, and the actual lack of transportation and storage of blood samples. But it will also be difficult to reproduce this scheme in other research centers.

The scientific achievement of the work is that, for the first time, the negative impact of low levels of NK cytotoxicity on the reproductive process was demonstrated, as well as the fact that with the level of growth of NK cytotoxicity, the negative impact increases. This can only be shown by large clinical groups.

## Figures and Tables

**Figure 1 diagnostics-12-00908-f001:**
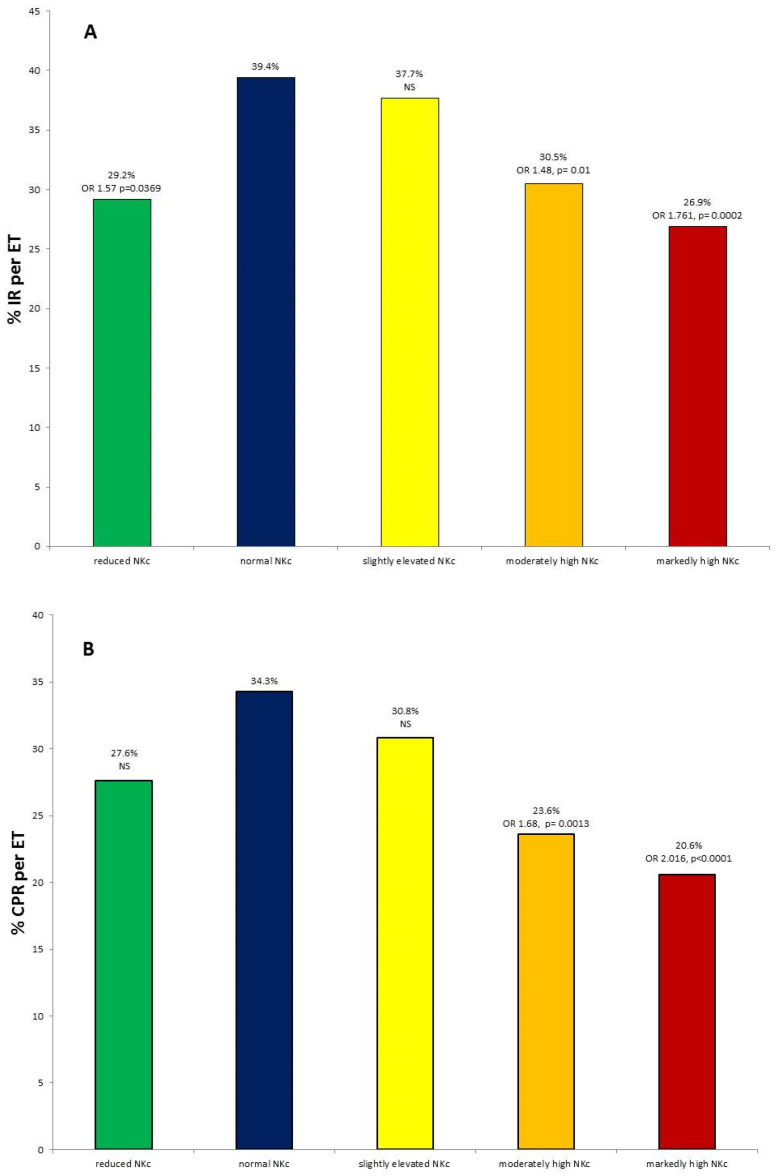
(**A**) Implantation rate (IR), (**B**) Clinical pregnancy rate (CPR) and (**C**) Live-birth-rates in patients with different NK cytotoxicity levels. IR, CPR andLBR was calculated per ET. (* *p* = 0.06 compared to patients with normal NKc. ** *p* < 0.05 compared to patients with normal NKc).

**Figure 2 diagnostics-12-00908-f002:**
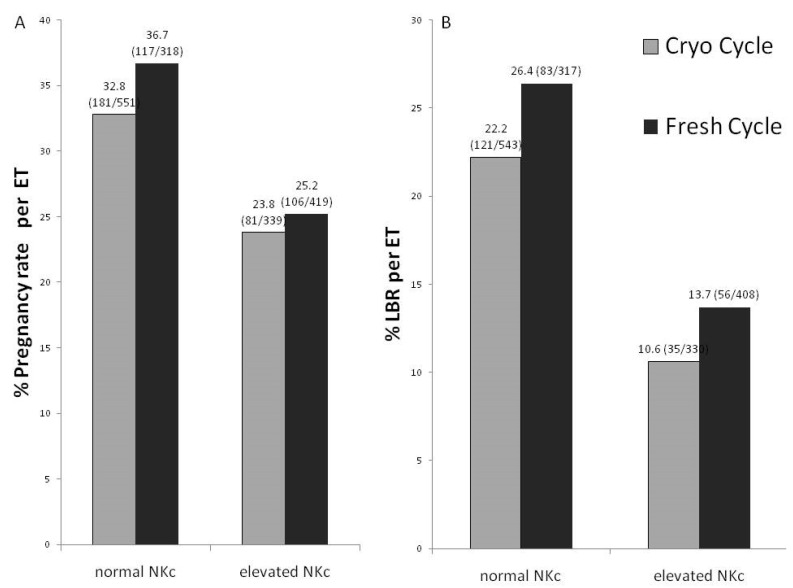
Clinical pregnancy rate (**A**) and live-birth rates (**B**) in patients with normal NKc and elevated NKc that have undergone fresh ET (in stimulated cycles) or cryo-ET in non-stimulated cycles. Both were calculated per ET and no significant difference was found.

**Table 1 diagnostics-12-00908-t001:** Clinical anamnesis and hormonal characteristic of randomized groups. *p* < 0.05 * compared to “Normal NKc: group ** *p* = 0.1 compared to “Normal NKc: group.

	Age	Number of Unsuccessful Programs	Endometrial Thickness (mm)	Number of Transferred Embryos	AMH (ng/mL) 2–3 Day of Cycle	FSH, (Mmo/Ml) 2–3 Day of Cycle	LH, (mMO/mL) 2–3 Day of Cycle
elevated NKc (*n* = 759)	35.4 ± 4.3	2.01 ± 0.8	10.78 ± 1.5	1.82 ± 0.3	2.7 ± 1.7 **	8.9 ± 3.5	7.2 ± 3.1
Normal NKc (*n* = 871)	35.0 ± 4.1	1.91 ± 0.6	10.96 ± 1.1	1.91 ± 0.2	3.2 ± 1.5	8.8 ± 3.0	6.7 ± 2.9
reduced NKc (*n* = 123)	33.6 ± 4.4	2.7 ± 0.4 *	10.6 ± 1.6	1.85 ± 0.3	2.90 ± 1.8	8.41 ± 3.7	7.5 ± 3.5
	No difference	*p* < 0.05 *	No difference	No difference	*p* = 0.1 **	No difference	No difference

**Table 2 diagnostics-12-00908-t002:** Clinical results of IVF cycles in patients with different NKc status (BcP- biochemical pregnancy).

	ET	IR	?* Subsequent information	BcP	Ectopic Pregnancy	CPR	?**	Anembryonic Pregnancy	Pregnancy Failure <12 w/g	Pregnancy Failure 12–24 w/g	Malformations	LBR	Caesarean Section
Elevated NKc	**759**	**233** **30.7%**	*1*	*40* *5.2%*	*6*	**187** **24.7%**	*20*	*13* *7.8%*	*55* *32.9%*	*5* *3%*	*0*	**91** **12.3%**	48
Normal NKc	**871**	**343** **39.4%**	*2*	*39* *4.4%*	*2*	**298** **34.3%**	*11*	*4* *1.4%*	*57* *19.8%*	*11* *3.8%*	*3*	**204** **23.8%**	119
Reduced NKc	**123**	**36** **29.35%**	*0*	*3* *2.4%*	*0*	**34** **27.6%**	*4*	*4* *1.3%*	*6* *20%*	*1* *3.3%*	*0*	**19** **15.9%**	8

?* no subsequent clinical information (* Excluded from CPR calculation, ** Excluded from LBR calculation).

**Table 3 diagnostics-12-00908-t003:** Success rates of IVF cycles in patients with different NKc status.

	IR	CPR	LBR	Pregnancy Failure @
Elevated NKc	30.7% (233/759) *	24.7% (187/758) *	12.3% (91/738) *	43.7% (73/167) *
Normal NKc	39.3% (343/871)	34.3% (298/869)	23.8% (204/858)	26.1% (75/287)
Reduced NKc	29.3% (36/123) *	*27.6% (34/123)*	*15.9% (19/119)* **	36.7 (11/30) *

* significant when compared to normal NKc group *p* < 0.05. ** not quite significant when compared to normal NKc group *p* < 0.07. @ all include—Anembryonic Pregnancy, Pregnancy failure <12 w/g, Pregnancy failure 12–24 w/g, and malformations.

## Data Availability

Data available on request due to privacy/ethical restrictions.

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
