# Peer review of "Accentuated Peripheral Blood NK Cytotoxicity Forms an Unfavorable Background for Embryo Implantation and Gestation"

_diagnostics, 2022, doi:10.3390/diagnostics12040908_

Round 1

Reviewer 1 Report

The authors investigate a possible correlation between elevated or reduced peripheral blood NK cytotoxicity and embryo implantation and gestation.

The introduction is well-written, providing sufficient background while including several relevant references for the reader.

In the Material and Methods, the authors describe clearly their protocols, study design, and statistical analysis.

In page 2, line 7, the authors mention the term “egg in uterus”, although the meaning is of course clear I would suggest to use a more appropriate term such as gestational sac or yolk sac depending what exactly they checked in the ultrasound.

In page 3, line 107, the authors should consider mentioning that K562 is an NK cell specific target line.

In page 3, line 126, the authors mention for the first time PBLs. In my experience PBL is a shortcut of peripheral blood lymphocytes (so T, B, and NK). So either there is a further isolation of lymphocytes from the PBMCs, or it was accidental and the authors meant to write PBMC. In any case, I would encourage the authors to clear this.

In page 3, line 124 and 128, (and also in page 9, lines 337-339) supplemental Fig1A and 1B are mentioned. However, these are not provided in the separate file that was provided (Non-published material). In the file that I received, exists only a supplementary table (Supplementary  table 1. A. Anamnestic characteristic in randomized groups (Pregnancy failures PF and ectopic pregnancies EP) B Live-Birth-Rate in patients with unsuccessful pregnancy’s in anamnesis.).

In page 4, lines 133-134, the authors write “As the concept of "normal cytotoxicity," the reference values of NK cytotoxicity favorable for reproductive prognosis was accepted in accordance with.6,17”. I assume they mean in accordance with the two references that they provide, but I find that the sentence ends a bit abrupt. The authors should consider changing it to i.e. “in accordance with our previous work. 6,17” or something similar.

In page 4, lines 137-138, the authors write, “In addition, the same "normal cytotoxicity levels" was defined as level were no correlation between NK% and NKc.18”. I find that something is missing grammatically form the sentence. Maybe “In addition, the same "normal cytotoxicity levels" was defined as the level were no correlation between NK% and NKc existed.18”?

The results are in their most part clearly presented and interpreted.

In table 2, page 4, I would encourage the authors to include not only the absolute numbers but also the percentages for “Anembryonic Pregnancy”, “Pregnancy failure <12w/g”, and “Pregnancy failure 12-24w/g”.

Throughout the results there are some slightly different terms used for the same thing. For example:

Implantation rate (IR) is clearly defined as a positive hCG (>50u/ml on the 14th day after ET) in page 2, lines 69-70. The acronym IR is used throughout the text, but on table 2 the authors write “Positive hCG test >50u/ml on the 14th day after ET. I believe it is simpler to also use IR here.

Clinical Pregnancy rate (CPR) is also clearly defined in page 2, line 74-75. However, in table 2 the authors write “Uterine pregnancy”, and on table 3 “PR”.

Finally, Live birth rate (LBR) is defined in page 2, line 77. However, in table 2 the authors write “childbirth”.

Of course, the meaning does not change, but in my opinion, it is better to stick to exactly the same terms in order to avoid confusion.

Throughout the results, there are some discrepancies in the numbers. For example:

  • Reduced NKc LBR is 16.0% in table 2, but 15.9% in figure 1
  • Normal NKc CPR is 34.3% in table 2, but 34.2% in page 5, line 178
  • Slightly elevated NKc LBR is 17.6% in figure 1 but 17.5% in page 5, line 177
  • Normal NKc LBR is 23.8% in table 2 and figure 1, but 23.4% in page 5, line 178
  • Moderately high NKc LBR is 14.2% in figure 1, but 14.3% in page 6, line 180
  • High NKc LBR is 6.9% in figure 1, but 6.8% in page 6, line 182

It is obvious to me that these are the same numbers (maybe rounded in a different way?), but I would encourage the authors to have a second look.

Concerning Figure 1, I would encourage the authors to consider including also the IR, and the CPR data in similar graphs. It could be for example, Fig.1A IR% per ET, Fig.1B CPR% per ET, Fig.1C LBR% per ET. This way the reader can have a visual aid for the data given in the text and the graphs can be a bit smaller and not so dramatic. This however, is a personal preference and the authors are of course free to visualise their data as they see fit.

Finally, the conclusions presented in the discussion are, in my opinion, supported both from the results and from the provided references.

In page 8, line 273, the authors should define the uRPL acronym.

In page 8, line 274, the authors should define and explain the uNK acronym. Does it refer to NK cells residing in the uterus/endometrium/decidua, or to a separate subpopulation of NK cells (CD56brightCD16low) which is sometimes referred to as uNK (irregardless of whether they are found in the uterus or in the periphery and in contrast to the CD56dimCD16high NK cells which are referred to as pNK cells).

In page 9, lines 309-310, the authors write, “NK and T cell numbers in peripheral blood do not correlate with endometrial NK and T frequency”. I would encourage the authors to cite a relevant publication (or publications) here. For example, I am aware of a 2011 publication from Laird SM et al. who compared CD56+ cells in peripheral blood and the endometrium and found no correlation. Has there been a similar finding for the T cells or other cell populations?

In page 9, line 316, the authors should again define and explain the pNK and dNK acronyms to avoid confusion.

Minor formatting or language concerns:

There are some double spaces throughout the text. For example: page 2, lines 75, 77, 92, 95; page 3, line: 102; page 4, line 131; page 7, lines 207, 209, 221, 222, 224, 234, 245; page 8, lines 259, 276, 293, 300, 301; page 9, lines 311, 318.

I believe a comma is needed on page 3, line130 after “After vortexing”, and on page 7, line 221 after “Markedly”.

On page 6, line 190, I would encourage the authors to use “barely”, “scarcely”, or “hardly” instead of “almost did not”.

On page 7, lines 208-209, the sentence “Only, patients with BMI (<20) had a not quite significantly higher frequency of accentuated elevated NKc than in normal BMI groups (20-30) (p=0.08 OR1.39).” is not very clear.

On page 7, lines 232, I would encourage the authors to change the word “Anyhow”. Personally, I consider this something quite informal that I would not use in writing.

On page 7, line 246, I would suggest to substitute the word “mother” with the “the maternal”

On page 9, line 307, I believe something is missing in the sentence. Maybe “Embryo implantation is regulated by fetal - endometrial NK recognition”?

On page 9, lines 330-332, the authors write, “NK cytotoxicity did not depend on age (in fertile period), anamnesis, reasons, duration, and infertility factors”. It is not clear what kind of reasons, and duration of what exactly?

In general, I have the impression that the text in the discussion is not always entirely clear concerning the language. Some parts are perfectly clear, and others have, in lack of a better term, a strange choice of words. For example in page 8, lines 269-270, “the inconvenience of implanting and pregnancy ongoing in patients with low levels of NKc”, or in page 9, lines 332-333, “Its disadvantage was realized equally during the stimulated cycle and 332 for the transfer of the frozen embryo”. 

Since I am not a native speaker myself, maybe I am of course wrong, but I would encourage the authors to have another look.

Overall, I found the manuscript very interesting to read.

Author Response

  1. Comments and Suggestions for Authors

Response for Reviewer 1

Thank you for your in-depth and thorough analysis of the manuscript

We have taken into account all the recommendations and made changes to the text.

-corrected a number of inaccuracies

-added information to the table and made fig1a 1b

-we also add an additional figure 1/2 which was not in the submission.

The article became better understood and informative

Thank you for your cooperation

The authors investigate a possible correlation between elevated or reduced peripheral blood NK cytotoxicity and embryo implantation and gestation. The introduction is well-written, providing sufficient background while including several relevant references for the reader.In the Material and Methods, the authors describe clearly their protocols, study design, and statistical analysis.In page 2, line 7, the authors mention the term “egg in uterus”, although the meaning is of course clear I would suggest to use a more appropriate term such as gestational sac or yolk sac depending what exactly they checked in the ultrasound.

  • We correct it

In page 3, line 107, the authors should consider mentioning that K562 is an NK cell specific target line.

We add information in text (NK-sensitive, HLA-negative. Chronic  myelogenous leukemia)

In page 3, line 126, the authors mention for the first time PBLs. In my experience PBL is a shortcut of peripheral blood lymphocytes (so T, B, and NK). So either there is a further isolation of lymphocytes from the PBMCs, or it was accidental and the authors meant to write PBMC. In any case, I would encourage the authors to clear this.

We add information in text

In page 3, line 124 and 128, (and also in page 9, lines 337-339) supplemental Fig1A and 1B are mentioned. However, these are not provided in the separate file that was provided (Non-published material). In the file that I received, exists only a supplementary table (Supplementary  table 1. A. Anamnestic characteristic in randomized groups (Pregnancy failures PF and ectopic pregnancies EP) B Live-Birth-Rate in patients with unsuccessful pregnancy’s in anamnesis.).

We add supplementary fig1 a b

In page 4, lines 133-134, the authors write “As the concept of "normal cytotoxicity," the reference values of NK cytotoxicity favorable for reproductive prognosis was accepted in accordance with.6,17”. I assume they mean in accordance with the two references that they provide, but I find that the sentence ends a bit abrupt. The authors should consider changing it to i.e. “in accordance with our previous work. 6,17” or something similar.

We add (in accordance with our previous work)

In page 4, lines 137-138, the authors write, “In addition, the same "normal cytotoxicity levels" was defined as level were no correlation between NK% and NKc.18”. I find that something is missing grammatically form the sentence. Maybe “In addition, the same "normal cytotoxicity levels" was defined as the level were no correlation between NK% and NKc existed.18”?

We correct it

The results are in their most part clearly presented and interpreted.

In table 2, page 4, I would encourage the authors to include not only the absolute numbers but also the percentages for “Anembryonic Pregnancy”, “Pregnancy failure <12w/g”, and “Pregnancy failure 12-24w/g”.

Throughout the results there are some slightly different terms used for the same thing. For example:

Implantation rate (IR) is clearly defined as a positive hCG (>50u/ml on the 14th day after ET) in page 2, lines 69-70. The acronym IR is used throughout the text, but on table 2 the authors write “Positive hCG test >50u/ml on the 14th day after ET. I believe it is simpler to also use IR here.

Clinical Pregnancy rate (CPR) is also clearly defined in page 2, line 74-75. However, in table 2 the authors write “Uterine pregnancy”, and on table 3 “PR”.

Finally, Live birth rate (LBR) is defined in page 2, line 77. However, in table 2 the authors write “childbirth”.

Of course, the meaning does not change, but in my opinion, it is better to stick to exactly the same terms in order to avoid confusion.

We correct according recommendations

Throughout the results, there are some discrepancies in the numbers. For example:

  • Reduced NKc LBR is 16.0% in table 2, but 9% in figure 1
  • Normal NKc CPR is 34.3% in table 2, but 34.2% in page 5, line 178
  • Slightly elevated NKc LBR is 17.6% in figure 1 but 17.5% in page 5, line 177
  • Normal NKc LBR is 23.8% in table 2 and figure 1, but 23.4% in page 5, line 178
  • Moderately high NKc LBR is 14.2% in figure 1, but 14.3% in page 6, line 180

High NKc LBR is 6.9% in figure 1, but 6.8% in page 6, line 182

 It is obvious to me that these are the same numbers (maybe rounded in a different way?), but I would encourage the authors to have a second look.

  • We correct all according  according to one approach to mathematical rounding of numbers (in text and fig/tab)

Concerning Figure 1, I would encourage the authors to consider including also the IR, and the CPR data in similar graphs. It could be for example, Fig.1A IR% per ET, Fig.1B CPR% per ET, Fig.1C LBR% per ET. This way the reader can have a visual aid for the data given in the text and the graphs can be a bit smaller and not so dramatic. This however, is a personal preference and the authors are of course free to visualise their data as they see fit.

We add 2 Fig with IR and CPR results

Finally, the conclusions presented in the discussion are, in my opinion, supported both from the results and from the provided references.

In page 8, line 273, the authors should define the uRPL acronym. 

We correct it

In page 8, line 274, the authors should define and explain the uNK acronym. Does it refer to NK cells residing in the uterus/endometrium/decidua, or to a separate subpopulation of NK cells (CD56brightCD16low) which is sometimes referred to as uNK (irregardless of whether they are found in the uterus or in the periphery and in contrast to the CD56dimCD16high NK cells which are referred to as pNK cells).

We add uterine NK

In page 9, lines 309-310, the authors write, “NK and T cell numbers in peripheral blood do not correlate with endometrial NK and T frequency”. I would encourage the authors to cite a relevant publication (or publications) here. For example, I am aware of a 2011 publication from Laird SM et al. who compared CD56+ cells in peripheral blood and the endometrium and found no correlation. Has there been a similar finding for the T cells or other cell populations?

We add reference in text

In page 9, line 316, the authors should again define and explain the pNK and dNK acronyms to avoid confusion.

We correct it

Minor formatting or language concerns:

There are some double spaces throughout the text. For example: page 2, lines 75, 77, 92, 95; page 3, line: 102; page 4, line 131; page 7, lines 207, 209, 221, 222, 224, 234, 245; page 8, lines 259, 276, 293, 300, 301; page 9, lines 311, 318.

I believe a comma is needed on page 3, line130 after “After vortexing”, and on page 7, line 221 after “Markedly”.

We add

On page 6, line 190, I would encourage the authors to use “barely”, “scarcely”, or “hardly” instead of “almost did not”.

We correct on hardly

On page 7, lines 208-209, the sentence “Only, patients with BMI (<20) had a not quite significantly higher frequency of accentuated elevated NKc than in normal BMI groups (20-30) (p=0.08 OR1.39).” is not very clear.

We correct it

On page 7, lines 232, I would encourage the authors to change the word “Anyhow”. Personally, I consider this something quite informal that I would not use in writing.

We correct it

On page 7, line 246, I would suggest to substitute the word “mother” with the “the maternal”

We correct it

On page 9, line 307, I believe something is missing in the sentence. Maybe “Embryo implantation is regulated by fetal - endometrial NK recognition”?

We correct it

On page 9, lines 330-332, the authors write, “NK cytotoxicity did not depend on age (in fertile period), anamnesis, reasons, duration, and infertility factors”. It is not clear what kind of reasons, and duration of what exactly?

We delete it. this phrase is not very successful and I think it is superfluous

In general, I have the impression that the text in the discussion is not always entirely clear concerning the language. Some parts are perfectly clear, and others have, in lack of a better term, a strange choice of words. For example in page 8, lines 269-270, “the inconvenience of implanting and pregnancy ongoing in patients with low levels of NKc”, or in page 9, lines 332-333, “Its disadvantage was realized equally during the stimulated cycle and 332 for the transfer of the frozen embryo”. 

We correct it

Since I am not a native speaker myself, maybe I am of course wrong, but I would encourage the authors to have another look.

Overall, I found the manuscript very interesting to read.

thanks

Especially with recommendations for corrections and improvements

Reviewer 2 Report

The manuscript by Boris Dons’koi et al. deals with the investigation of  using the NKc for routine diagnostic in IVF patients. In particular, the Authors perform cytotoxicity test of NKc 17 to K562 was measured in PBMC (peripheral blood mononuclear cells) by flow cytometry before ET. 

The novelty of this study is limited, in view of the fact that several research groups have previously investigated, in deeper detail and by diversified techniques, similar aspects as those described in the present manuscript.

In the Introduction, immunosuppressive mechanisms involved in the dampening/inhibition of NK cell functions should be included in more detail and the corresponding references added. The description of the current knowledge about NK cells in pregnancy and in pathological condition should be explained.

References: their choice is not always appropriate and should be replaced by new one. Moreover, additional references regarding recent important studies on the characterization of NK cells in decidual tissues and in peripheral blood should be included.

Discussion: Authors should better highlight the novelty of their findings and, especially, discuss their results in the context of the broad existing literature on the same topic.

English language and style need revision: some grammar mistakes should be amended and several sentences should be rephrased

Author Response

2  Comments and Suggestions for Authors

Response for Reviewer 2

The manuscript by Boris Dons’koi et al. deals with the investigation of using the NKc for routine diagnostic in IVF patients. In particular, the Authors perform cytotoxicity test of NKc 17 to K562 was measured in PBMC (peripheral blood mononuclear cells) by flow cytometry before ET. 

The novelty of this study is limited, in view of the fact that several research groups have previously investigated, in deeper detail and by diversified techniques, similar aspects as those described in the present manuscript.

So I agree that this problem (NK for reproduction) has been researched and published for more than 20 years. However, the results are quite ambiguous. Thus, the main scientific value of our study is not the topic itself, but the large array of ordinary populations that has been studied. According to scheme- 1 protocol of NKc investigation / 1 IVF-clinic and clinData collection, this study is the largest of all. In addition, it is not a specialized group of patients with> 3 or> 5 reproductive failures. but the usual IVF population and the usual routine NK testing. We added some explanations to the text.

In the Introduction, immunosuppressive mechanisms involved in the dampening/inhibition of NK cell functions should be included in more detail and the corresponding references added. The description of the current knowledge about NK cells in pregnancy and in pathological condition should be explained.

We added some explanations to the text. In this study we did not touch on the issue of treatment. it is even more complex and ambiguous. We are currently preparing an article about this.

References: their choice is not always appropriate and should be replaced by new one. Moreover, additional references regarding recent important studies on the characterization of NK cells in decidual tissues and in peripheral blood should be included.

We add more appropriate and new one.  We add more information and Ref  about  decidual/endometrial and pbNK relationship. 

Discussion: Authors should better highlight the novelty of their findings and, especially, discuss their results in the context of the broad existing literature on the same topic.

We added some explanations to the text.

English language and style need revision: some grammar mistakes should be amended and several sentences should be rephrased

We conducted additional proofreading of the text and correct more than 50 corrections of grammar mistakes and rephrased sentences.

Thank you for your in-depth and thorough analysis of the manuscript